# The Asylum Seekers in Non-Metropolitan Areas in France: Between Temporary Integration and Leading to Autonomy. The Case of the Ambertois Territory

**Rafik Arfaoui** 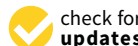

Université Clermont Auvergne, AgroParisTech, Inra, Irstea, VetAgro Sup, Territoires,
F-63000 Clermont–Ferrand, France; mohammed_rafik.arfaoui@uca.fr

**Abstract:** This article focuses on the integration process of people seeking asylum in non-metropolitan areas in France. It conceptualizes the reception of asylum seekers involving two interrelated approaches: the utilitarian approach and the humanitarian approach. This article is based on surveys, participatory and sensitive cartography, and participant observation conducted in the Ambertois territory between 2017 and 2018. I find the Ambertois territory can be considered a "fragile space," particularly in terms of demographics, with difficulties in maintaining public services. These difficulties are risks for asylum seekers, and are impacting the urban space. These risks are intensified by the national and regional level policies like the recent reform of the asylum and immigration act on the one hand, and the suffering they experienced throughout their migratory journey on the other. Faced with these risks, local synergies, which facilitate the integration of asylum seekers, are emerging from local actors. This integration is temporary and is considered by local actors as leading to the autonomy of asylum seekers.

**Keywords:** asylum seekers; non-metropolitan areas; fragile spaces; temporary integration; autonomy; dispersal policy; France

---

## 1. Introduction

The Ambertois territory within the Puy-de-Dôme in the Auvergne-Rhône-Alpes region provides an example of the settlement of asylum seekers in central France (see Figure 1). This settlement operation has been conducted since July 2016. In this operation, four municipalities (Ambert, Arlanc, Cunlhat, and Saint-Amant-Roche-Savine) coordinated together to create a reception center for asylum seekers. The settlement operation is part of a wider national dispersal policy for migrants conducted in France since July 2015.[1] Some of these reception centers have been created in non-metropolitan areas because of their relatively low cost compared to other, more expensive parts of France. However, while the housing availability and cost is lower in these non-metropolitan areas, the commitment of local actors to host asylum seekers on their territory may not be as strong as metropolitan areas. The reception center for asylum seekers in the Ambertois territory contains apartments located in different sites in the four municipalities. The lives of the asylum seekers I spoke with were punctuated by administrative and medical appointments, encounters with neighbors, and feelings of solitude.

---

[1] The dispersal policy refers here to the asylum reform introduced in France on 29 July 2015. It aims to resettle migrants, who live in Paris and around Calais in particular, to other parts of France, through the National Reception System (*DIspositif National d'Accueil*). Thus, asylum seekers cannot decide where they will stay. The French state decides where they will be housed. For more information: https://www.immigration.interieur.gouv.fr/Asile/La-reforme-de-l-asile-issue-de-la-loi-du-29-juillet-2015/Loi-n-2015-925-du-29-juillet-2015-relative-a-la-reforme-du-droit-d-asile.

Encounters with neighbors allowed them to escape the solitude of their living arrangements. Volunteers provided activities that responded to some of the needs of asylum seekers and reduced the risks they faced from a restrictive asylum policy and the "fragile space" that Ambertois provides for integration.

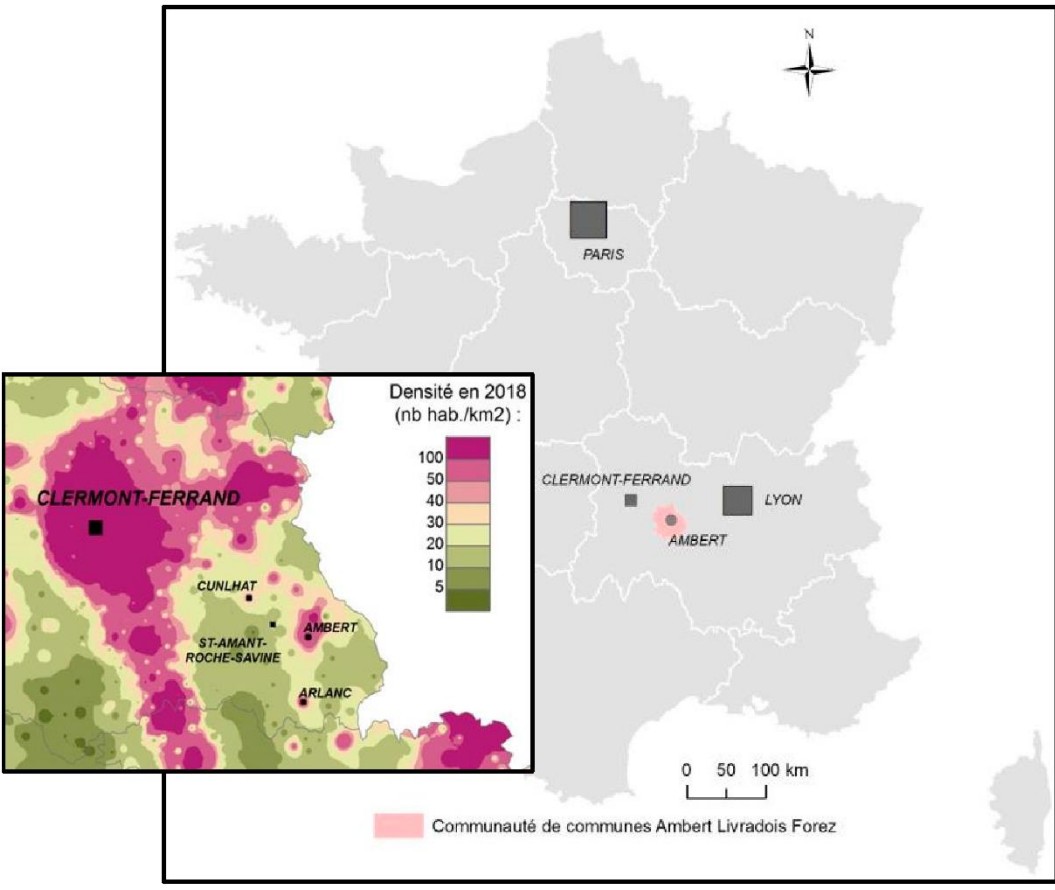

**Figure 1.** The Ambertois territory. Source: Rafik Arfaoui, INSEE data base, 2018.

In this article, I will first explain the dispersal policy of migrants in France and the resettlement system for asylum seekers, particularly in non-metropolitan areas. Following this, I will present how the Ambertois territory can be considered a fragile space. Finally, I will discuss the urban impact of asylum seekers, and their experiences in the Ambertois territory. The analysis of urban impacts reveals two interrelated approaches in the Ambertois territory: the utilitarian approach and the humanitarian approach. The analysis of asylum seekers' experiences reveals that their integration involves developing autonomy by building local connections that respond to real risks.

## 2. Methods

My research focused on the reception of asylum seekers in non-metropolitan areas. I chose this area because the geography literature on asylum seekers in France has often focused on large cities, but little research has been done on non-metropolitan areas (CAMIGRI 2016). Since 2015, non-metropolitan areas have become increasingly important in migration studies. In addition to my thesis, I can also mention the "Camigri" project (*Les campagnes françaises dans la dynamique des migrations internationales*), which aims to analyze the settlement of migrants in rural areas in France.

This article is based on field surveys conducted between 2017 and 2018. First, I identified the actors involved in the reception and integration of asylum seekers: Institutional and associative actors,[2] volunteers and displaced persons' support groups, and asylum seekers themselves (See Figure 2, Tables 1 and 2). For my key informant interviews, I used semi-structured interviews, each lasted between 45 min and 2 h.

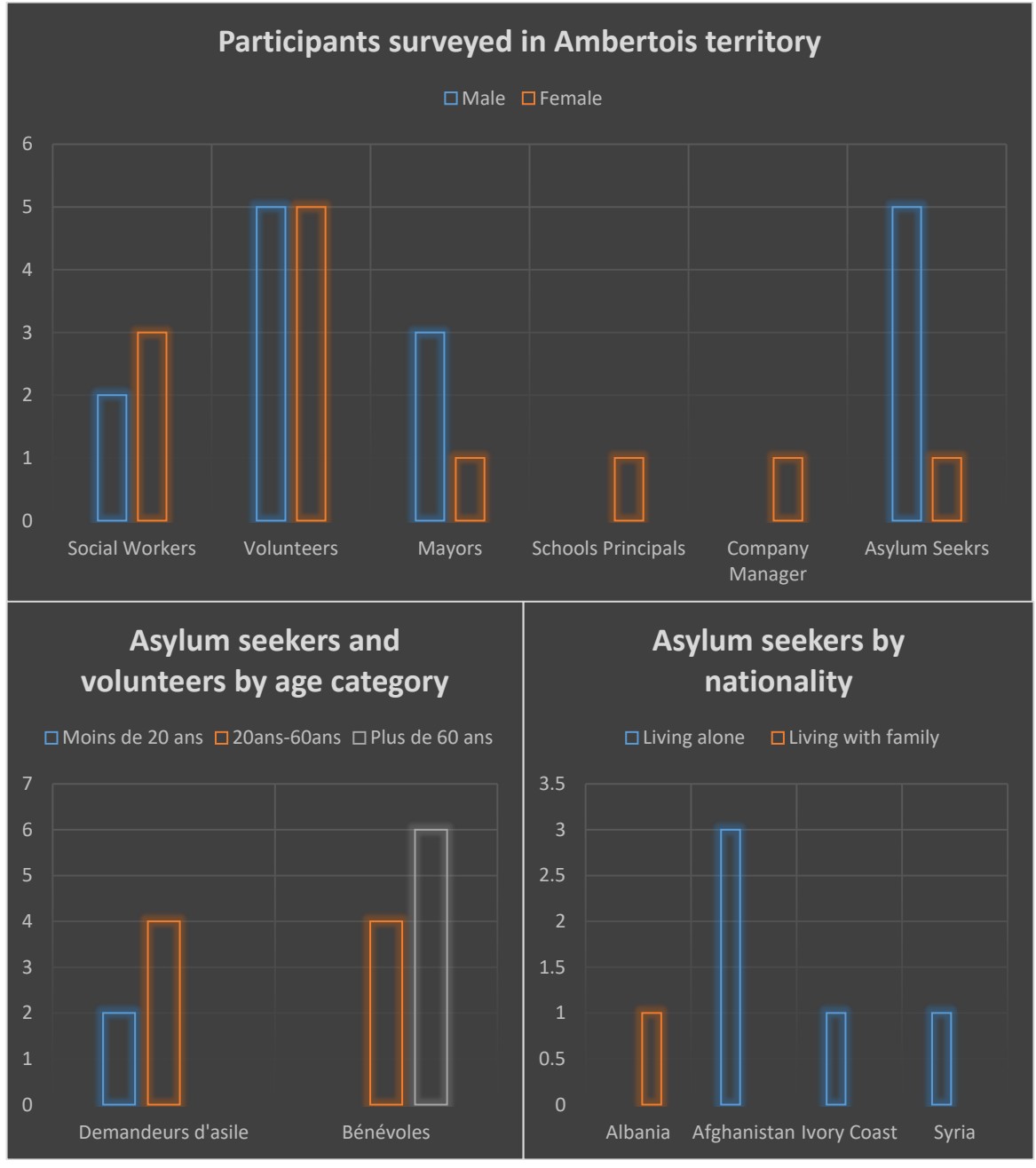

**Figure 2.** The participants surveyed in the Ambertois territory. Source: Rafik Arfaoui, field research data base, 2017 to 2018.

---

2   I define institutional actors as mayors and governmental officials. Associative actors include the directors of the reception center for asylum seekers, humanitarian organizations, and social workers.

**Table 1.** The participants by their title, category, and institution. Source: Rafik Arfaoui, field research data base, 2017 to 2018.

| Institution | Category | Title of Participant |
|---|---|---|
| Elementary School | Educational Institution | Principal |
| Samwill Company | Industrial Company | Manager |
| Municipality of Ambert | Municipal Government | Councilor' Municipality |
| Municipality of Arlanc | Municipal Government | Mayor |
| Municipality of Cunlhat | Municipal Government | Mayor |
| Municipality of Saint-Amant-Roche-Savine | Municipal Government | Mayor |
| *Collectif l'Élégante* | Non-governmental Organization | Volunteers |
| Not related in any Institution | | Volunteers |
| Reception center for asylum seekers | Humanitarian Organization | Social Workers |

**Table 2.** The asylum seekers surveyed by age, sex, and family situation. Source: Rafik Arfaoui, field research data base, 2017 to 2018.

| Nationality | Age | Sex | Family Situation |
|---|---|---|---|
| Ivory Coast | 19 | Female | Living alone |
| Albania | 40 | Male | Living with 2 children and his wife |
| Afghanistan | 27 | Male | Living alone |
| Afghanistan | 20 | Male | Living alone |
| Afghanistan | 30 | Male | Living alone |
| Syria | 29 | Male | Living alone |

I chose to use participatory and sensitive cartography with participants because it offered a method for narrative decentralization (Mekdjian et al. 2014). Forms of visual expression, and not only discursive, appear through "cartographic gestures" (Mekdjian and Olmedo 2016). In other words, cartography goes beyond the constraints of the face-to-face discursive engagement. It conveys forms of symbolic violence, produced in particular by the administrations in charge of examining asylum applications in France. To accept an asylum application, these administrations interrogate asylum seekers about their life story in order to find elements to justify protection. This is done in a context of suspicion of "fake refugees." Our approach was partly of a reflection on the researcher's ethics, and a rethinking of the cartographic production method in geography by moving from a top–down model to a bottom–up model.

To this end, I organized cartography workshops involving asylum seekers living in the Ambertois territory. There were six participants, including one woman, aged between 19 and 40 years old. They were not paid for their participation. Contact with the participants was established through the coordinator of the reception center where they were housed. The participants spoke Arabic, French, English, and Dari. My ability to speak three of the four languages built a climate of trust and facilitated the exchange. For those who spoke Dari, one of the Afghan participants who also spoke English acted as a translator.

I explained to the participants that I do not belong to the reception center administration, but that I am a PhD student at the University of Clermont Auvergne. I introduced myself as an Algerian immigrant who came to France to study geography. This reinforced the climate of trust. At the same time, it raised questions of injustice (Young 1990) related to inequalities in freedom of movement and residence according to the administrative status of migrants. All these aspects were taken into consideration in the analysis of the maps produced by the participants.

The first step of the workshop was to co-create a key of their journey (see Figure 3). This key translated the co-production of a collective narrative about migration from individual experiences. To do this, first, all participants attached sticky notes that described their experiences of the trip. Then, participants grouped these sticky notes and created keyword categories. Finally, a visual representation was created using stickers in different colors and shapes.

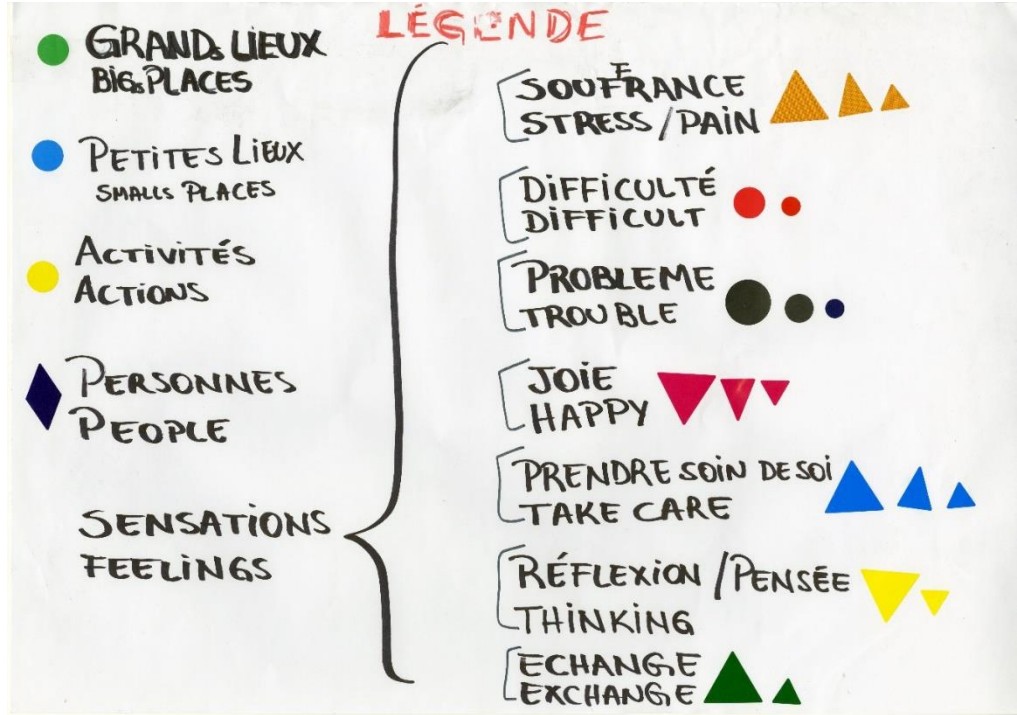

**Figure 3.** Collective key on the exile experience of asylum seekers hosted in the Ambertois territory. Source: Rafik Arfaoui and participants of workshops, July to August 2017.

## 3. Background of Reception of Asylum Seekers

Since 2015, the French government has been using a new asylum and immigration act, which aims to disperse exiles from the concentration areas (Paris and the area around Calais) to other parts of the country. Land opportunities and the relatively low cost of reception, and the desire of local politicians, have placed non-metropolitan areas at the heart of the national dispersal policy. For this reason, many reception centers have been created. This dispersal policy is multi scalar and it can be found at European, national, and local levels (see Figure 4).

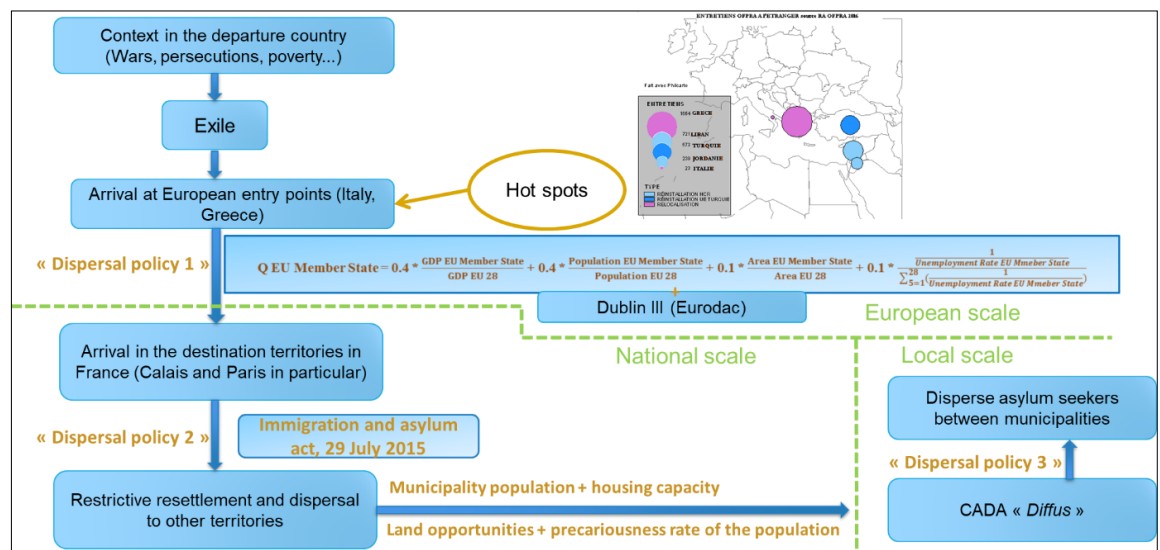

**Figure 4.** Multi-scalar dispersal of exiles' policy. Source: Knowledge of migration policies, Rafik Arfaoui, 2017.

### 3.1. On Ambertois Territory

Once they have arrived in the reception areas, people seeking asylum are, in some cases, dispersed many times. Since 2016, a new category of accommodation for asylum seekers has been introduced, known as a diffused reception center. The idea is that people are hosted on different sites and often in several municipalities. Several municipalities then network with the aim to create a "fair" distribution of the number of asylum seekers. This is the case in the Ambertois territory where asylum seekers are resettled in four municipalities: Ambert, Arlanc, Cunlhat, and Saint-Amant-Roche-Savine. This suggests that inhabitants, institutional actors, and associative actors have little perspective on the issue of reception and integration. This temporality dimension has been taken into account in my analysis.

### 3.2. On France

Since the end of the 18th century, France has proclaimed the constitutional right of asylum. In article 120 of the 1793 Constitution, "the French people give asylum to foreigners banished from their homeland for the cause of freedom. He denies it to tyrants." Even if it will not be applied, this article places the reflection on asylum in France in a tradition that goes back more than two centuries. Since the 1951 Geneva Convention and the 1969 New York Protocol, France has granted asylum to persons considered "at risk" in their country of origin. The protection rates of asylum seekers have changed significantly since the 1970s. Indeed, the OFPRA (Office Français de Protection des Réfugiés et des Apatrides), which granted refugee status to 85% of exiles seeking asylum in 1973, refused it to 85% of them in 1990.

The French State introduced a new immigration and asylum act on 29 July 2015. The areas known as "migrant camps", particularly in Paris and Calais, have been highly publicized. The government's intention was, in addition to protecting the exiles, to "make them invisible." To this aim, persons wishing to be accommodated in reception centers for asylum seekers no longer had the possibility to choose their residence area. They were forced to accept the proposed accommodation at the risk of losing their right to the Asylum Seekers' Allowance. Thus, several asylum seekers found themselves in areas they had not chosen. The creation of reception centers had to meet four criteria: The size of the municipality population, accommodation capacity, land opportunities, and the precariousness of the population.

### 3.3. On the EU

On a European scale, there are dispersal policies through so-called "relocation" operations. These operations aim to distribute people seeking asylum from hot spots (Greece, Turkey, Jordan, and Lebanon). These hot spots aim to identify people seeking asylum and to remove undesirables (Agier 2008). This eviction is part of an approach to asylum that distinguishes "real" refugees from "fake" refugees. The distribution policy from hot spots is based on the observation that each European country must take its share of "responsibility" for the reception of asylum seekers. This distribution is a reminder of the measure proposed by Jean-Claude Junker to "distribute" asylum seekers among the 28 EU Member States. Four indicators were taken into consideration to calculate each country's quota: Population size, country area, gross domestic product (GDP), and unemployment rate (Schneider et al. 2013). Another example of the dispersal policy in the EU is the Dublin II and III regulations. These regulations imply that asylum seekers must do so in the country where their fingerprints were first taken. Verifications are carried out via the Eurodac data system. If an EU State, where the person is seeking asylum, finds that a fingerprint has previously been taken in another EU country, then a removal procedure is initiated to that country. People are identified as "Dubliners". This forced displacement measure is produced in a spirit of "burden sharing". The idea of burden refers more broadly to a perception of exiles by host States as a "burden" to be supported. Following this analysis, two questions became important: Has the process of resettlement brought something positive? Has it made facilitated the integration of asylum seekers?

## 4. Data and Findings

### 4.1. The Ambertois Territory: A Fragile Space

Fragile space is a theme that appeared in human geography around 1985. Like other themes that emerged before or after this period, such as *"espaces défavorisés"*[3] or *"territoires sensibles,"*[4] it describes situations of "crisis" facing spaces. For the definition of fragile spaces, Laurent Rieutort (2006) underlines the idea of zones *"prêtes à se briser"*[5] (Rieutort 2006, p. 15). The concept of fragile space is therefore associated with the dimension of the risk of worsening the situation of a space. It may be related to its demographic, social, and economic dynamics in relation to national and regional averages or to an average specific to the space object under study (Edouard 2017).

First, there are demographic indicators. The statistical indicators concerning the average annual rate of change in the population between 2010 and 2015 highlight that the four municipalities where the reception center is located were losing population (see Table 3). Of course, the dynamics are more complex when we look at the details of this demographic evolution, between migratory changes and the rate of natural increase. However, there needs to be a minimum number of people within a geographic area to keep public facilities (schools, hospitals, etc.) functioning. Next, there are socio-economic indicators. The municipality of Saint-Amant-Roche-Savine lost jobs between 2010 and 2015. Saint-Amant-Roche-Savine is in a more critical situation in that it has a higher unemployment rate than the rest of the municipalities, or even compared to the regional and national averages (see Table 3).

**Table 3.** Statistical elements on demographic and economic fragility in the Ambertois territory. Source: INSEE data base, 2018.

| Name of the Municipality | Average Annual Rate of Change in Population Growth in % (2010–2015) | Population Growth Due to the Natural Increase in % (2010–2015) | Population Growth Due to Net Migration in % (2010–2015) | Change in Total Employment: Average Annual Rate between in % (2010–2015) | Unemployment Rate of 15–64 Year Olds in % (2015) |
|---|---|---|---|---|---|
| **Ambert** | −0.5 | −0.5 | 0.0 | +0.2 | 10.9 |
| **Arlanc** | −0.4 | −0.7 | +0.4 | +0.8 | 14.8 |
| **Cunlhat** | −0.7 | −1.9 | +1.2 | +0.3 | 11.7 |
| **Saint-Amant-Roche-Savine** | −1.3 | −0.9 | −0.3 | −6.9 | 21.9 |
| **Auvergne-Rhône-Alpes Region** | +0.8 | +0.4 | +0.3 | +0.3 | 12.2 |
| **France (metropolitan)** | +0.5 | +0.4 | +0.1 | 0.0 | 13.7 |

### 4.2. The Local Opinions on the Reception of Asylum Seekers in the Ambertois Territory

In addition to the statistics on a fragile space, what is local public opinion on the reception of asylum seekers in the Ambertois territory? Of particular interest is the case of Arlanc, which registered 43.21% of the votes for Marine Le Pen, the candidate of the "Front National" (FN) party, in the second round of the 2017 presidential elections. It is a nationalist party that has historically placed the "immigration problem" or "national preference" at the heart of its electoral agendas. To collect local public opinion, I focused on the few local press articles that talk about this subject.

---

[3]    Translation: Disadvantaged areas.
[4]    Translation: Sensitive areas.
[5]    Translation: Ready to be broken.

While immigration has been identified by some inhabitants as a possible explanation of the relative importance of the vote for the FN in 2017, others highlight that it may be due to loss of public services or to the media effect.[6] Gilles Charreyron's (2015) analysis of the FN vote in Auvergne highlights the progressive anchoring of the party in the east of Auvergne, more industrialized, since the 2007 and 2012 presidential elections. *"Les populations de ces territoires industriels de l'est auvergnat sont davantage aux prises avec une crise économique et une mondialisation qui créent de l'incertitude (désindustrialisation, délocalisations des entreprises, dumping social, perte de l'identité collective). La crainte de l'incident dans la vie professionnelle et sociale (chômage partiel, plans sociaux, reconversion professionnelle...) et la peur du déclassement font naître une inquiétude latente qui se retrouve probablement dans les urnes. Des électeurs fragilisés adhèrent plus facilement à la rhétorique du Front national sur le repli sur soi, la nécessaire protection des frontières face à la mondialisation et à la construction européenne, le « problème de l'immigration » et la « préférence nationale » en matière d'emploi et d'allocations."* (Charreyron 2015, p. 11)[7]. This analysis, including the example of Arlanc, illustrates that the relative increase for the FN vote is not directly related to the settlement of asylum seekers, since it occurred only in 2016, four years after the 2012 election, taken into consideration in Gilles Charreyron's analysis (Charreyron 2015). It is mainly related to economic problems and the loss of public services.

## 5. Discussion: Utilitarian and Humanitarian Approaches

Two main approaches structure the reception of people seeking asylum: The humanistic approach and the utilitarian approach. These two approaches refer to a current of psychology for the first, and a concept of political philosophy for the second, which is not going to be detailed here. By humanistic approach, in the context of my research, I mean any action on the part of one or more actors who, in their commitment to the reception of persons seeking asylum, act for humanitarian reasons. As for the utilitarian approach, these actors place the positive economic and demographic contribution resulting from the reception of people seeking asylum for the areas at the heart of their strategy. When, for example, an actor refers to the "duty of reception" in relation to humanitarian emergency, we call this vision a humanistic approach. When in other interviews, actors highlight that "the reception of people seeking asylum is good for the economy and the demography of the areas," we call this vision, a utilitarian approach. There may be, in the discourse of some actors, a mix of these two approaches.

### 5.1. Utilitarian Approach

In non-metropolitan areas characterized by elements of demographic (loss of population) and/or economic fragility (unfilled jobs and loss of jobs), the utilitarian approach is present in the reception strategy according to the profile of the actors. During analysis, two important examples of the utilitarian approach of welcoming were visible in the Ambertois territory. First, there was the fight against housing vacancies in the social housing stock. With demographic difficulties and facing problems of attractiveness of the population, several social housing initiatives have remained vacant for several years. In Cunlhat, a building owned by a social landlord *"Ophis du Puy-de-Dôme,"* has been vacant for more than five years (see Figure 5). The installation of the reception center has mobilized vacant housing to accommodate asylum seekers. For the social landlord, this makes it possible to rent vacant housing again and avoid financial losses. It also allows the social landlord to advertise to the local government that it is committed to a priority issue for the state, the reception of asylum seekers. This

---

6    https://www.lamontagne.fr/arlanc-63220/politique/pourquoi-arlanc-vote-t-elle-massivement-front-national_12394910/#refresh.

7    Translation: "The populations of these industrial territories in eastern Auvergne are more affected by an economic crisis and globalization that create uncertainty (deindustrialization, relocation of companies, social dumping, loss of collective identity). The fear of the incident in professional and social life (partial unemployment, social plans, professional retraining...) and the fear of downgrading give rise to a latent anxiety that is probably reflected in the vote. Weakened voters more easily adhere to the rhetoric of the National Front on self-doubt, the need to protect borders in the face of globalization and European integration, the "immigration problem" and the "national preference" for employment and benefits".

commitment is used to assert itself with the state in future housing projects in which local government is an important actor. For mayors, mobilizing housing from the social landlord is seen as an opportunity to encourage them to develop other housing projects in their municipality.

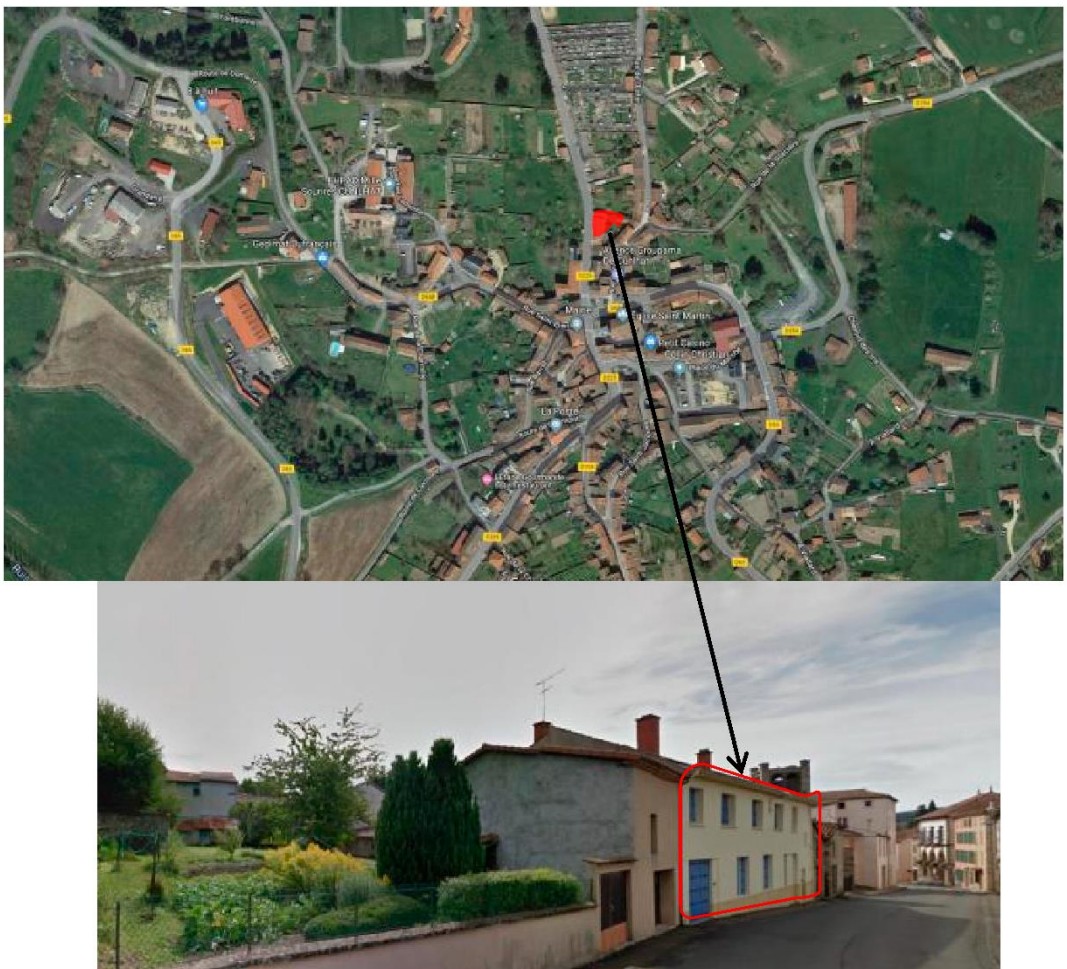

**Figure 5.** Vacant housing re-marketed to accommodate asylum seekers in Cunlhat. Base imagery © Google maps street view, 2019.

Furthermore, there is the example of school classes threatened with closure in areas with low population density. The elementary school in Saint-Amant-Roche-Savine had a problem in ensuring a required number of elementary school students to keep classes open. The mayor of the municipality requested, during the installation of the reception center, that his municipality welcome families with children. In addition to the availability of adequate housing for families with children in the municipality, this would make it possible to maintain the nursery school classes opened by the admission of asylum-seeking children. The approach of local actors, adopted in this context, was to focus on a duty to welcome asylum seekers.

In this context, the exiles' dispersal policy has made it possible to maintain public services and revitalize the housing market, severely affected by the demographic decline. For some interviewed volunteers, this observation is paradoxical, almost schizophrenic. On the one hand, the state decides to close public services because of the population decline; yet, on the other, the state decides to settle asylum seekers in these same territories with the risk that they will not have access to public services.

*5.2. Humanitarian Approach*

Other actors, most of them volunteers, choose a reception strategy that only assists people they consider vulnerable. For some of these volunteers, their approach to welcoming is part of a political demand to claim the rights of people considered vulnerable.

The rejected asylum seekers, without the right to housing or work, depend on the help provided by humanitarian associations and volunteer networks. The installation of the reception center for asylum seekers was accompanied by a significant outpouring of solidarity from part of the local population. The volunteers were involved in several tasks, including learning French, food aid, socio-cultural support, organizing walks, and accompanying people to administrative or medical appointments. Noting the risks faced by the first people on their territory who were rejected, measures were put in place by volunteers. Thus, groups of volunteers such as that of "l'Élégante", in the Ambertois territory, host two families, totaling 10 people, whose asylum requests have been rejected. This accommodation is in a building rented for this purpose and now called "*Résidence l'Élégante*" (see Figure 6).

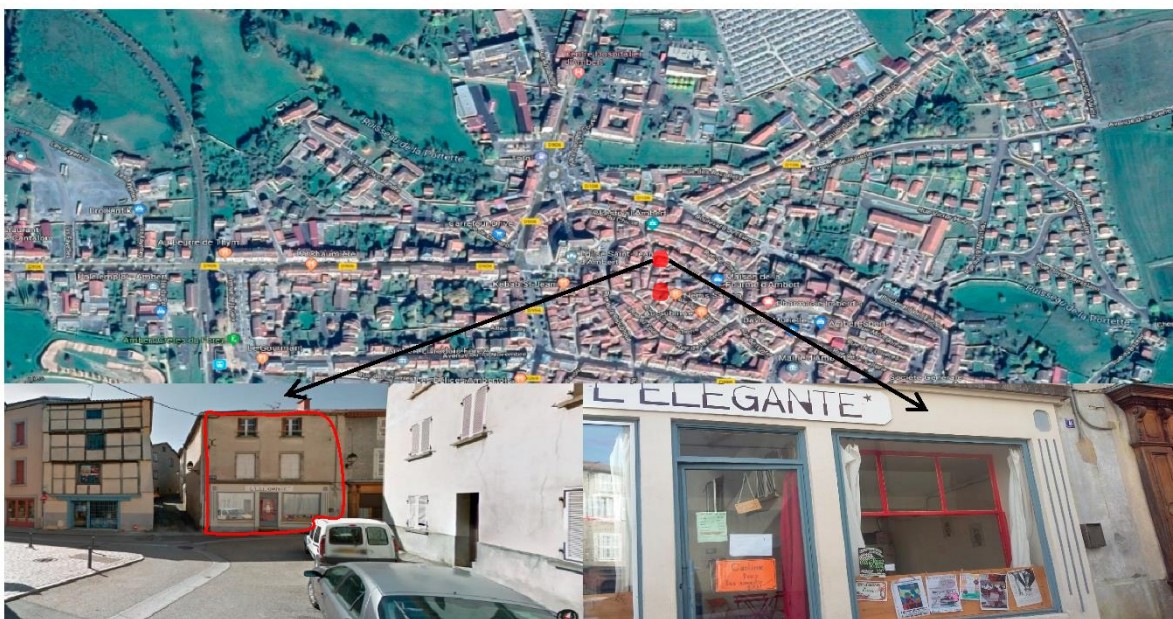

**Figure 6.** Residence "L'Élégante" in Ambert. Base imagery © Google maps street view, 2019.

In addition to accommodation, open-price canteens have been organized almost twice a week. In addition, there are musical events organized in the area with open-price admission. These activities have been organized in order to finance rental costs and survival of the families who have been refused accommodation. The "*Résidence l'Elégante*" also hosts spaces for political and cultural debates (feminism, installation of wind turbines, etc.). Some volunteers involved in the "*Collectif l'Élégante*" define themselves as anarchists. Some of them lived in large urban areas before settling in Ambert. Their actions are part of a struggle for social justice. The residents "L'Élégante" are not on the margins of this solidarity movement. They are involved through cooking, helping to organize cultural events, or even by proposing to participate in political debates. "I would like to organize a session to discuss the problems in my country. I have the impression that people here don't really know what's going on there." The approach adopted in this context calls for a right to reception.

The two humanistic and utilitarian approaches are interrelated and do not contradict one another. They reveal claims both for a duty to receive and a right to receive. This shift from the duty of some to the rights of others, has been reinforced with the exit of the first people to have their asylum claims rejected, in order to respond to the impasses of discretionary migration policies (absence of the right to exist at a place because of the absence of a residence permit issued by the State administration). For Agier (2018) in his book "*L'étranger qui vient: repenser l'hospitalité*", which is inspired by the writings of

Emmanuel Kant (1795) in his essay "*Projet de paix perpétuelle*" on movement from the duty of one to the right of the other would consist in transposing the ideal of universal hospitality, in the name of which a growing number of citizens are mobilized, into a rule of law that every foreigner has the right not to be treated as an enemy, as Kant put it" (Agier 2018, p. 142).

How do these two humanitarian and utilitarian approaches to reception affect the integration of asylum seekers in the Ambertois territory? In the next section, I will analyze the asylum seekers' experiences in the Ambertois territory.

## 6. Discussion: Asylum Seekers' Experiences

The practice of space by asylum seekers is linked to their administrative status. Without the right to work, to train, and to live in a place of their choice, asylum seekers practice reception spaces as retention spaces. This is truer when it concerns a welcome in an area with limited public transport access and limited access to the services available in metropolitan area, like in Clermont-Ferrand. Asylum seekers also practice reception areas as spaces of solidarity. Faced with the confinement imposed by administrative measures restricting some of their rights, volunteers and collectives supporting exiles help them to overcome the political, economic, social, and mobility constraints they face.

The maps drawn in the cartography workshops highlight these spaces experienced by asylum seekers as spaces of both detention and solidarity. Thus, Besnik, from Albania, seeking asylum and hosted in Ambert, experience his host territory between the sociability and the permanent stress of rejected his asylum application (see Figure 7). Qais, from Afghanistan, seeking asylum and hosted in Arlanc, experience his host territory as waiting space, symbolized by "office" and "coffee," and the need of a better tomorrow symbolized "go to work" (see Figure 8).

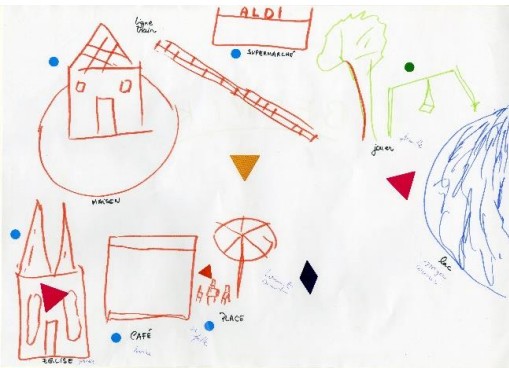

**Figure 7.** Map drawn by Besnik: Ambertois territory between stress and sociability. Source: Rafik Arfaoui, 2017.

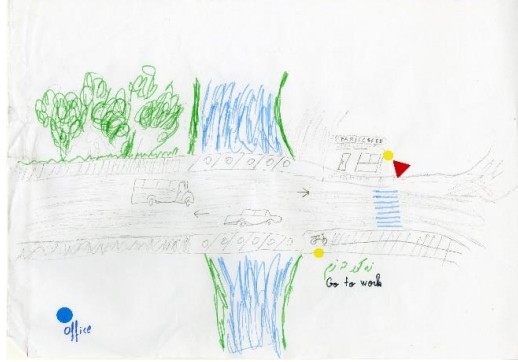

**Figure 8.** Map drawn by Qais: Ambertois territory between waiting and the desire to work. Source: Rafik Arfaoui, 2017.

Between detention and solidarity, reception areas become, for asylum seekers, intermediate space–time that separates them between the experiences lived during their exile, and their uncertain futures. This intermediate space–time is similar to the airlock of a home or cinema. It is used to protect the interior space from the nuisances that could be caused by the exterior space. In this airlock, asylum seekers experience a situation of permanent stress from an uncertain future. They are perceived by the State as potentially "fake" refugees, a "burden" to be wary of and to be controlled.

The intervention of actors supporting asylum seekers in this airlock aims to integrate them into local society. This integration is temporary until a response is received from the administrations in charge of treating their request. Temporary integration is the action of local actors in support of exiles to accompany them towards autonomy. The aim is to facilitate the acquisition of resources that would enable them to overcome their dependence on the asylum administration. My observation of temporary integration in the Ambertois territory is in line with Emmanuelle Bonderandi's (2008) assessment that the dynamics of temporary integration is a characteristic of the migration process in which people seeking asylum are enrolling.

The "Stevenson model" on "well-treating organizations" (Bardonnet et al. 2016) seems interesting to identify and classify the actions of local actors that lead to the integration of asylum seekers. The model proposes two definitions (assumptions) of well-treatment and abuse:

- Any act/sign that contributes to a sense of increased autonomy (feeling that their ability) is well treated.
- Any act/sign that contributes to a weakened sense of autonomy (feeling that their ability to maintain their balance has weakened) is abuse.

This model contrasts with Maslow's pyramid in the sense that psychological and physical needs are considered equal. Applied to the question of integration, this model implies that the action of local actors is carried out without a hierarchy of importance between psychological and physical needs. In other words, reception is no longer limited to physiological needs but goes beyond them. Based on my field surveys, I have identified and classified the synergies of local actors for the integration of asylum seekers in the Ambertois territory according to the "Stevenson model" (see Figure 9).

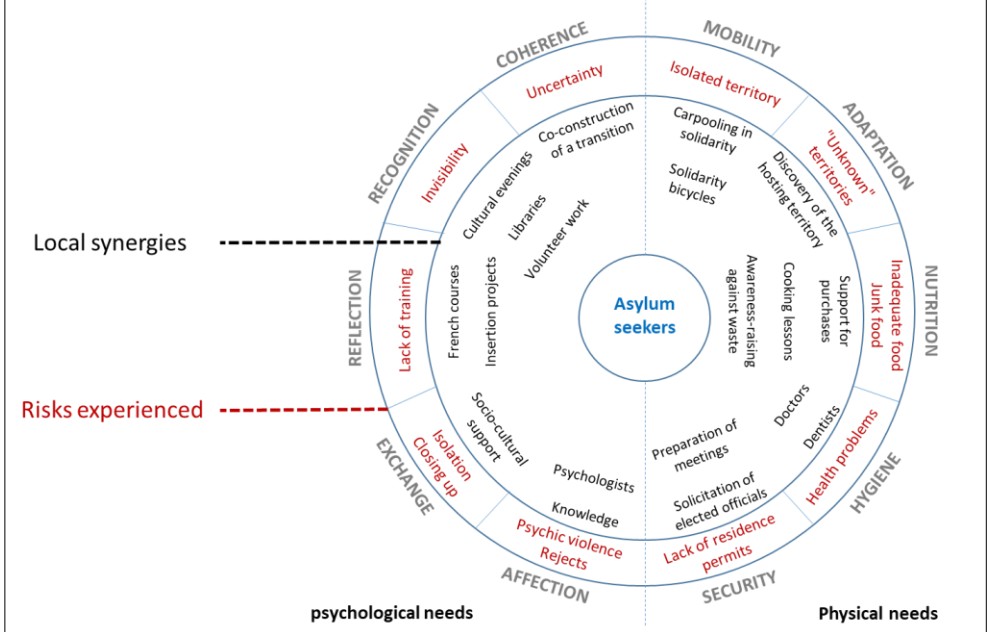

**Figure 9.** "Stevenson's model" applied to the integration of asylum seekers in the Ambertois territory. Source: Field research data base, Rafik Arfaoui, 2017 to 2018.

Asylum policy in France does not sufficiently consider the integration of asylum seekers. Unlike Germany, for example, the asylum reform of 29 July 2015 does not include a budget for French language training for asylum seekers, considered as the main tool for integration. Thus, we can see that asylum seekers learn the French language through workshops organized by volunteers. These workshops are conceived as spaces of sociability between volunteers and asylum seekers. The dispersal policy complicates the integration of asylum seekers when they are received in fragile areas with little access to public services and public transport. The role of volunteers and support groups for exiles plays a fundamental role here too, by creating local synergies aimed at providing the tools for the autonomy of asylum seekers.

## 7. Conclusions

The integration of asylum seekers in the Ambertois territory is caught in a conflict between the local and national levels. It was born from the observation of local actors that there is a need for asylum seekers to face experienced risks. On the one hand, these risks are produced by asylum policies in France, which lead to a social, economic, and cultural fragility of people seeking asylum. On the other, these risks result from a geographical reality in the Ambertois territory with elements of demographic and socio-economic fragility.

Focusing my research on people seeking asylum makes possible the identification of the temporal dimension of integration. This is temporary and responds to time scales specific to the asylum application in France. This temporary integration is meant to be a step in the exile process that will enable people seeking asylum to acquire the social and cultural tools that will lead to their autonomy. This autonomy goes beyond the spatial limits of the Ambertois territory and concerns the scale of the French territory. It is the political and social expression of the rejection of the State policy of confinement of asylum seekers.

This study returns to the question of the effectiveness of the new system for resettlement of asylum seekers in France. This study does not provide a concrete answer as to the appropriate relocation system. However, it does provide some suggestions. These suggestions should be complemented by other studies in other regions of France.

First, human resources must be adapted to the geographical context of the settlement area. For example, diffuse accommodation structures located in several municipalities involve frequent movements of social workers. It is necessary that the budget allocated to accommodation facilities for asylum seekers takes this aspect into account. In this regard, several social workers have denounced the progressive loss of the dimension of socio-cultural support for asylum seekers to administrative support.

Moreover, it is necessary that the resettlement of asylum seekers be accompanied by the development of public transport and public services that can respond to the needs of both local residents and asylum seekers. It is also important that asylum seekers can have access to an employment contract, when it has been possible, and benefit from a budget for language learning. These two points are fundamental to their integration.

Finally, on a more global scale, it is important to respect the right of human beings to move freely. There is a worldwide injustice regarding the mobility of people. While citizens of rich countries have the opportunity to access a large part of the world without a visa requirement, citizens of poor countries have access to a tiny part of the world and, at the same time, struggle to obtain visas that allow them to move freely. From the global to the local level, we are today seeing a ghettoization of the world where undesirables are reduced to circulating in restricted spaces. Today, the question is about the right of people to mobility.

**Funding:** This research was funded by [Université Clermont Auvergne] through my Doctoral Contract.

**Acknowledgments:** I would like to thank all those who have enabled me to do this work, especially those who are seeking asylum, social workers, volunteers and all the local actors I met during my research field in the Ambertois territory. I would also like to thank the reviewers who helped me to improve my article.

**Conflicts of Interest:** The author declares no conflict of interest.

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
