# Peer review of "The Asylum Seekers in Non-Metropolitan Areas in France: Between Temporary Integration and Leading to Autonomy. The Case of the Ambertois Territory"

_socsci, doi:10.3390/socsci8070210_

Round 1

Reviewer 1 Report

After review of the manuscript, I wish to congratulate the author(s).  The manuscript is innovative and very, very informative.  I have studied immigration and refugee/asylee policy for three decades and, even with that, I felt I learned a lot by reading the manuscript.  I have no question that the manuscript and its contents contribute to the scholarly literature focused on immigration and refugee/asylee policy.  I also thought the graphics were really well done and helped the reader understand fully what was the research focused on and how the research methods were employed--particularly interesting was the use of "participatory and sensitive mapping," and the use of the concept of "space." 

Author Response

Dear colleague,

Thank you for your analysis and your congratulations.

Reviewer 2 Report

Dear colleagues!

The plan of the article is excellent, but, unfortunately, the author(s) haven’t covered the topic thoroughly.

In my opinion, the value and the purposes of the new policy of spreading of applicants for political shelter throughout the whole country to avoid their concentration in particular city and their suburbs and emergence of some negative effects of ghettoisation. In general, the motives of the rural communities which accept such applicants are well disclosed. Even the process of integration (lines 247-257) is shown a little bit.

But the article lacks the main thing: the analysis of efficiency of integration process (however it is described as "temporary" in relation to this category of immigrants) in such small villages. How does it affect their mastering the French language?

The author interviewed the immigrants, but, unfortunately, did not share their subjective impressions of the integration process with us (it might have strengthened the work).

It would be extremely interesting to learn how not only mayors, social workers and volunteers perceive applicants for political shelter, but also get to know some public opinion of such small villages in general. It would be possible to learn the opinion of people by means of polls, the analysis of local newswires and social networks. It would be especially curious, taking into account the fact that 43% of voters voted for nationalist Marine Le Pen in the small village called Arlanc in the 2nd round of presidential elections in 2017.

But the most important issue is the following: a question, which touches some comparison of a new way of settling of the applicants for shelter with the former one, is left alone when their main capacity is concentrated in the cities. Have the process of resettlement brought something positive? Have it made facilitated the integration of immigrants?

Author Response

Dear colleagues,

Thank you for your comments and questions.

Language learning is strongly impacted by the asylum reform of 29 July 2015. There is no budget for training those seeking asylum in France, unlike in Germany for example. Language learning is therefore achieved through workshops organized by volunteers. The effectiveness of language learning varies according to the level of education of asylum seekers and the competence of volunteers to provide French language training.

The dispersal policy maintained public services and helped to reduce housing vacancies in the "Ambertois territory". However, it has negative repercussions on the lives of asylum seekers who arrive in a territory they have not chosen and characterised in particular by elements of demographic and socio-economic fragility and difficulties in accessing public transport. The integration of asylum seekers is negatively impacted in this case.

Round 2

Reviewer 2 Report

Dear colleagues!

I received answers to a number of the questions which interested me. The article was improved. It had some completeness of the presented material.

But I did not receive the answer to the main issue: whether the new system of resettlement of applicants of asylum seekers is better than the old one?

Just fancy that  I am a big Parisian politician or the official (the prime minister, the minister, the deputy minister, the Chairman of the Committee of the National Assembly). I should understand: how effective is the new system? Is it necessary to change it someway? Or should it be rejected absolutely?

The old system was not efficient because it promoted a ghettoization of migrants in city suburbs. I address to the ppesent scientific article to get the answer as the article has been prepared on the basis of interesting local investigation. The author of article points to a number of negative effects of a new system of resettlement in fragile rural areas, but he does not give the definite answer on the question which interests me.

Is it necessary to change something in the system of resettlement used since 2016? If it is necessary, what is it concrete? Or is it necessary to return to a former system? Or to touch nothing? Or can the author offer some third system? Or have we got few data so far to draw such conclusions and it is necessary to continue investigation further or to get the results from other regions of the country and from other types of settlements? That is what would be desirable to learn. In my opinion, it would be a logical end of the article.

Besides, the surname of the leader of the French nationalists is written incorrectly. Le Pen is the correct variant.

Author Response

Dear Colleague!

Thank you very much for your comments and pertinent questions. They allowed me to improve my article and enrich my study. Below, my answer to your question on the effectiveness of the system for resettlement of asylum seekers in France. I included this answer in the text at the conclusion level that I would like you to read.

Best regards,

Is the new system of resettlement of applicants of asylum seekers is better than the old one?

Response:

This study returns to the question of the effectiveness of the new system for resettlement of asylum seekers in France. This study does not provide a concrete answer as to the appropriate relocation system. However, it does provide some suggestions. These suggestions should be complemented by other studies in other regions of France.
First, human resources must be adapted to the geographical context of the settlement area. For example, diffuse accommodation structures located in several municipalities involve frequent movements of the social workers. It is necessary that the budget allocated to accommodation facilities for asylum seekers take this aspect into account. In the continuity of this reflection, several social workers denounce the progressive loss of the dimension of socio-cultural support for asylum seekers to administrative support. Then, it is necessary that the resettlement of asylum seekers be accompanied by the development of public transport and public services that can respond to the needs of both local residents and asylum seekers. It is also important that asylum seekers can have access to an employment contract when it's be possible and benefit from a budget for language learning. These two points are fundamental to their integration. Finally, on a more global scale, it is important to respect the right of human beings to move freely. There is a worldwide injustice regarding the mobility of people. While citizens of rich countries have the opportunity to access a large part of the world without a visa requirement, citizens of poor countries have access to a tiny part of the world and at the same time struggle to obtain visas that allow them to move freely. From the global to the local level, we are today seeing a ghettoization of the world where undesirables are reduced to circulating in restricted spaces. Today, the question is about the right of people to mobility.